# Detecting fabrication in large-scale molecular omics data

**Michael S. Bradshaw**[1]*, **Samuel H. Payne**[2]

**1** Computer Science Department, University of Colorado Boulder, Boulder, Colorado, United States of America, **2** Biology Department, Brigham Young University, Provo, Utah, United States of America

* michael.bradshawiii@colorado.edu

**Data Availability Statement:** All data used in this paper can be found in GitHub (https://github.com/MSBradshaw/FakeData).

**Funding:** This work was supported by the National Cancer Institute (NCI) CPTAC award [U24 CA210972] awarded to SP. The funders had no

## Abstract

Fraud is a pervasive problem and can occur as fabrication, falsification, plagiarism, or theft. The scientific community is not exempt from this universal problem and several studies have recently been caught manipulating or fabricating data. Current measures to prevent and deter scientific misconduct come in the form of the peer-review process and on-site clinical trial auditors. As recent advances in high-throughput omics technologies have moved biology into the realm of big-data, fraud detection methods must be updated for sophisticated computational fraud. In the financial sector, machine learning and digit-frequencies are successfully used to detect fraud. Drawing from these sources, we develop methods of fabrication detection in biomedical research and show that machine learning can be used to detect fraud in large-scale omic experiments. Using the gene copy-number data as input, machine learning models correctly predicted fraud with 58–100% accuracy. With digit frequency as input features, the models detected fraud with 82%-100% accuracy. All of the data and analysis scripts used in this project are available at https://github.com/MSBradshaw/FakeData.

## Introduction

Fraud is a pervasive problem and can occur as fabrication, falsification, plagiarism or theft. Examples of fraud are found in virtually every field, such as: education, commerce, and technology. With the rise of electronic crimes, specific criminal justice and regulatory bodies have been formed to detect sophisticated fraud, creating an arms-race between methods to deceive and methods to detect deception. The scientific community is not exempt from the universal problem of fraud, and several studies have recently been caught manipulating or fabricating data [1, 2] or are suspected of it [3]. More than two million scientific articles are published yearly and ~2% of authors admit to data fabrication [4]. When these same authors were asked if they personally knew of colleagues that had fabricated, falsified, or modified data, positive response rates rose 14–19% [4, 5]. Some domains or locales have somewhat higher rates of data fabrication; in a recent survey of researchers at Chinese hospitals, 7.37% of researchers admitted to fabricating data [6]. Overall, these rates of data fabrication potentially mean tens to hundreds of thousands of articles are published each year with manipulated data.

role in study design, data collection and analysis, decision to publish, or preparation of the manuscript. https://proteomics.cancer.gov/programs/cptac.

**Competing interests:** The authors have declared that no competing interests exist.

Data in the biological sciences is particularly vulnerable to fraud given its size—which makes it easier to hide data manipulation—and researcher's dependence on freely available public data. Recent advances in high-throughput omics technologies have moved biology into the realm of big-data. Many diseases are now characterized in populations, with thousands of individuals characterized for cancer [7], diabetes [8], bone strength [9], and health care services for the general populace [10]. Large-scale characterization studies are also done for cell lines and drug responses [11, 12]. With the rise of importance of these large datasets, it becomes imperative that they remain free of errors both unintentional and intentional [13].

Current methods for ensuring the validity of research is largely limited to the peer-review process which, as of late, has proven to be insufficient at spotting blatant duplication of images [14], let alone subtleties hidden in large scale data. Data for clinical trials can be subject to reviews and central monitoring [15, 16]. However, the decision regarding oversight methodology and frequency is not driven by empirical data, but rather is determined by clinics' usual practice [17]. The emerging data deluge challenges the effectiveness of traditional auditing practices to detect fraud, and several studies have suggested addressing the issue with improved centralized and independent statistical monitoring [5, 6, 16, 18]. However, these recommendations are given chiefly to help ensure the safety and efficacy of the study, not data integrity.

In 1937, physicist Frank Benford observed in a compilation of 20,000 numbers that the first digit did not follow a uniform distribution as one may anticipate [19]. Instead, what Benford observed was that digit 1 occurred about 30% of the time, 2–18%, 3–13%, and that the pattern continues decaying, ending with digit 9 occurring < 5% of the time. Why this numerical pattern exists can be explained by looking at the relative change from lower vs higher first digit numbers. For example, moving a value from 1,000 to 2,000 is a 100% increase, while changing from 8,000 to 9,000 is an only increase of 12.5%. This pattern holds true in most large collections of numbers, including scientific data, where the upper and lower limits are not tightly bound. Comparing a distribution of first digits to a Benford distribution can be used to identify deviations from the expected frequency, often because of fraud. Recently Benford's law has been used to identify fraud in financial records of international trade [20] and money laundering [21]. It has also been used on a smaller scale to reaffirm suspicions of fraud in clinical trials [3]. It should be noted that Benford's Law, despite being called a law, it not always followed and does have some limitations. If the upper and lower limits of a dataset are tightly bound (the dataset cannot span orders of magnitudes of values), a Benford-law like digit distribution may not be able to form.

The distinction between fraud and honest error is important to make; fraud is the intent to cheat [5]. This is the definition used throughout this paper. An honest error might be forgetting to include a few samples, while intentionally excluding samples would be fraud. Incorrectly copying and pasting values from one table to another is an honest error, but intentionally changing the values is fraud. In these examples the results may be the same but the intent behind them differs wildly. In efforts to maintain data integrity, identifying the intent of the misconduct may be impossible and is also a secondary consideration after suspect data has been identified.

Data fabrication is "making up data or results and recording or reporting them" [5]. This type of data manipulation when not documented for bonafide applications such as simulation or imputation of missing values is free from the above ambiguity relating to the author's intent. Making up data "such that the research is not accurately represented in the research record" [5] is always wrong. We explore methods of data fabrication and detection in molecular omics data using supervised machine learning and Benford-like digit-frequencies. We do not attempt to explain why someone may choose to fabricate their data as other study have done [6, 22];

our only goal is to evaluate the utility of digit-frequencies to differentiate real from fake data. The data used in this study comes from the Clinical Proteomic Tumor Analysis Consortium (CPTAC) cohort for endometrial carcinoma, which contains copy number alteration (CNA) measurements from 100 tumor samples [23, 24]. We created 50 additional fake samples for these datasets. Three different methods of varying sophistication are used for fabrication: random number generation, resampling with replacement, and imputation. We show that machine learning and digit-frequencies can be used to detect fraud with near perfect accuracy.

## Methods

### Real data

The real data used in this publication originated from the genomic analysis of uterine endometrial cancer. As part of the Clinical Proteomics Tumor Analysis Consortium (CPTAC), 100 tumor samples underwent whole genome and whole exome sequencing and subsequent copy number analysis. We used the results of the copy number analysis *as is*, which is stored in our GitHub repository at https://github.com/MSBradshaw/FakeData.

### Fake data

Fake data used in this study was generated using three different methods. In each method, we created 50 fake samples which were combined with the 100 real samples to form a mixed dataset. The first method to generate fake data was random number generation. For every gene locus, we first find the maximum and minimum values observed in the original data. A new sample is then fabricated by randomly picking a value within this gene specific range. The second method to generate fake data was sampling with replacement. For this, we create lists of all observed values across the cohort for each gene. A fake sample is created by randomly sampling from these lists with replacement. The third method to generate fake data is imputation performed using the R package missForrest [25], which we repurposed for data fabrication. A fake sample was generated by first creating a copy of a real sample. Then we iteratively nullified 10% of the data in each sample and imputed these NAs with missForrest until every value had been imputed and the fake sample no longer shared any data originally copied from the real sample (S1 Fig).

### Machine learning training

With a mixed dataset containing 100 real samples and 50 fake samples, we proceeded to create and evaluate machine learning models which predict whether a sample is real or fabricated (S2 Fig). The 100 real and 50 fake samples were both randomly split in half, one portion added to a training set and the other held out for testing. Given that simulations on biological data like this have never, to our knowledge, been done, we did not have any expectation as to which type of model would perform best at this task. Thus, we tried a wide variety of models, all implementing fundamentally different algorithms. Sticking to models included in SciKit Learn [26] with a common interface increased code reusability and allowed for quick and consistent comparison. Using Python's SciKitLearn library, we evaluated five machine learning models:

1. Gradient boosting (GBC): [27] an ensemble method based on the creation of many weak decision trees (shallow trees, sometimes containing only 2 leaf nodes).

2. Naïve Bayes (NB): [28] type of probabilistic classifier based on Bayes Theorem.

3. Random Forest (RF): [29] ensemble method of many decision trees, differs from GBC in that the decision trees are not weak, they are full trees working on slightly different subsets of the training features.

4. K-Nearest Neighbor (KNN): [30] this does not perform any learning per-se, but classifies based on proximity to labeled training data.

5. Support Vector Machine (SVM): [31] is a statistical based learning method that operates by trying to maximize the size of the gap between classification categories.

Training validation was done using 10-fold cross validation. We note explicitly that the training routine was never able to use testing data. After all training was complete, the held-out test set was then fed to each model for prediction and scoring. We used simple accuracy and F1 scores as evaluation metrics. For each sample in the test set, ML models would predict whether it was real or fabricated. Model accuracy was calculated as the number of correct predictions divided by the number of total predictions. To assess the amount of false positives and false negatives we also compute the F1 score [32]. The entire process of fake data generation and ML training/testing was repeated 50 times. Different random seeds were used when generating each set of fake data. Thus, fake samples in all 50 iterations are distinct from each other. Grid search parameter optimization was performed to select the hyperparameter set for each model. The parameter search spaces used, and all of the data and analysis scripts used in this project, are available at https://github.com/MSBradshaw/FakeData. A full list of the final parameters used for each model-dataset pair can be found in "S1 File".

We compared two types of input to the machine learning models here. For the first we use gene copy-number data from CPTAC as the features (17,156 training features/genes in total) with added fabricated samples as the training and test data. In the second section, rather than directly using the copy-number values, we use the proportional frequency of the digits 0–9 in the first and second positions after the decimal place (digit-frequencies). This results in 20 training features in total, those features being: frequency of 0 in the first position, frequency of 1 in the first position . . . frequency of 9 in the first position, frequency of 0 in the second position, frequency of 1 in the second position. . . frequency of 9 in the second position.

## Benford-like digit frequencies

Benford's Law or the first digit law has been instrumental at catching fraud in various financial situations [20, 21] and in small scale clinical trials [3]. The distribution of digit frequencies in a set of numbers conforming to Benford's Law has a long right-tail; the lower the digit the greater its frequency of occurrence. The CNA data used here follows a similar pattern (S3 Fig). The method presented here is designed with the potential to generalize and be applied to multiple sets of data of varying types and configurations (i.e. different measured variables (features) and different quantities of variables). Once trained, machine learning models are restricted to data that conform to the model input specifications (i.e. the same number of input features, for example). Converting all measured variables to digit frequencies circumvents this problem. Digit frequencies are calculated as the number of occurrences of a single digit (0–9) divided by the total number of features. In the method described in this paper, a sample's features are all converted to digit frequencies of the first and second digit after the decimal. Thus for each sample the features are converted from 17,156 copy number alterations to 20 digit frequencies. Using this approach, whether a sample has 100 or 17,156 features it can still be trained on and classified by the same model (though it's effectiveness will still be dependent on the existence of digit-frequency patterns).

## Computing environment

Data fabrication was performed using the R programming language version 3.6.1. For general computing, data manipulation, and file input output we used several packages from the tidy-verse: [33] *readr*, *tibble*, and *dplyr*. Most figures were generated using *ggplot2* in R, with *grid*, and *gridExtra* filling some gaps in plotting needs. Data fabricated with imputation was performed using the *missForest* package [25].

The machine learning aspect of this study was performed in Python 3.8.5. All models and methods for the evaluation used came from the package SciKit-Learn (sklearn) version 0.23.2 [26]. Pandas version 1.1.3 was used for all reading and writing of files [34]. The complete list of parameters used for each model and dataset pair can be found in the supplemental material online, "S1 File".

## Results

Our goal is to explore the ability of machine learning methods to identify fabricated data hidden within large datasets. Our results do not focus on the motivations to fabricate data, nor do they explore in depth the infinite methodological ways to do so. Our study focuses on whether machine learning can be trained to correctly identify fabricated data. Our general workflow is to take real data and mix in fabricated data. When training, the machine learning model is given access to the label (i.e. real or fabricated); the model is tested or evaluated by predicting the label of data which was held back from training (see Methods).

### Fake data

The real data used in this study comes from the Clinical Proteomic Tumor Analysis Consortium (CPTAC) cohort for endometrial carcinoma, specifically the copy number alteration (CNA) data. The form of this real data is a large table of floating point values. Rows represent individual tumor samples and columns represent genes; values in the cells are thus the copy number quantification for a single gene in an individual tumor sample. This real data was paired with fabricated data and used as an input to machine learning classification models (see Methods). Three different methods of data fabrication were used in this study: random number generation, resampling with replacement, and imputation (S1 Fig). The three methods represent three potential ways that an unscrupulous scientist might fabricate data. Each method has benefits and disadvantages, with imputation being both the most sophisticated and also the most computationally intense and complex. As seen in Fig 1, the random data clusters are far from the real data. Both the resampled and imputed data cluster tightly with the real data in a PCA plot, with the imputed data also generating a few reasonable outlier samples.

To look further into the fabricated data, we plotted the distribution of the first two digits after the decimal place in the real and fake data (S3 Fig). While none of the fake have quite the spread of digit distributions in terms of variation, data created via imputation matches the real data the closest in terms of mean digit frequencies. We also examined whether fake data preserved correlative relationships present in the original data (S4 Fig). This is exemplified by two pairs of genes. PLEKHN1 and HES4 are adjacent genes found on chromosome 1p36 separated by ~30,000 bp. Because they are so closely located on the chromosome, it is expected that most copy number events like large scale duplications and deletions would include both genes. As expected, their CNA data has a Spearman correlation coefficient of 1.0 in the original data, a perfect correlation. The second pair of genes, DFFB and OR4F5, are also on chromosome 1, but are separated by 3.8 Mbp. As somewhat closely located genes, we would expect a modest correlation between CNA measurements, but not as highly correlated as the adjacent gene pair. Consistent with this expectation, their CNA data has a Spearman correlation coefficient

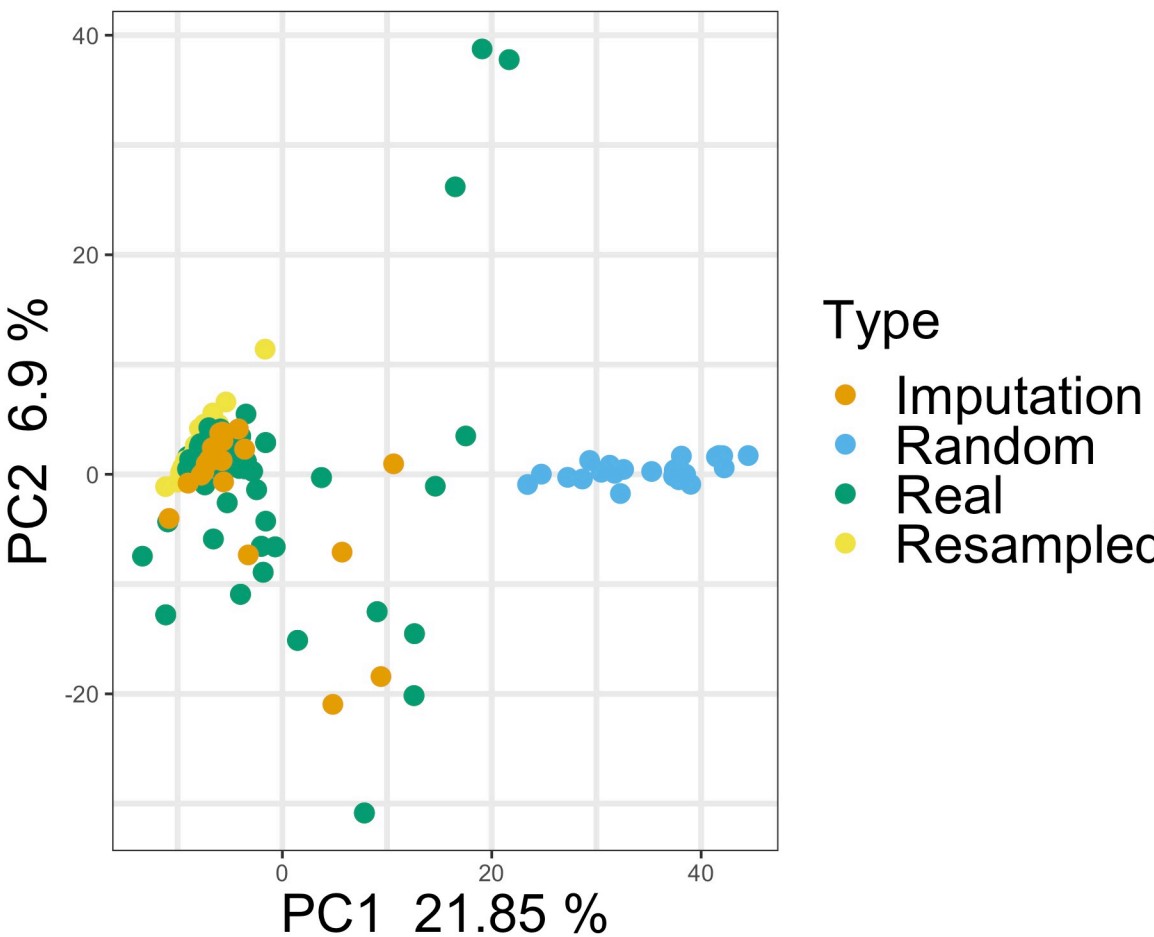

**Fig 1. Principal component analysis of real and fake samples.** Copy number data for the real and fabricated samples are shown. The fabricated data created via random number generation is clearly distinct from all other data. Fabricated data created via resampling or imputation appears to cluster very closely with the real data.

of 0.27. Depending on the method of fabrication, fake data for these two gene pairs may preserve these correlative relationships. When we look at the random and resampled data for these two genes, all correlation is lost (S4C–S4F Fig). Imputation, however, produces data that closely matches the original correlations, PLEKHN1 and HES4 $R^2$ = 0.97; DFFB and OR4F5 $R^2$ = 0.32 (S4G and S4H Fig).

### Machine learning with quantitative data

We tested five different methods for machine learning to create a model capable of detecting fabricated data: Gradient Boosting (GBC), Naïve Bayes (NB), Random Forest (RF), K-Nearest Neighbor (KNN) and Support Vector Machine (SVM). Models were given as features the quantitative data table containing copy number data on 75 labeled samples, 50 real and 25 fake. In the copy number data, each sample had measurements for 17,156 genes, meaning that each sample had 17,156 features. After training, the model was asked to classify held-out testing data containing 75 samples, 50 real and 25 fake. The classification task considers each sample separately, meaning that the declaration of real or fake is made only from data of a single sample. We evaluated the models on accuracy (Fig 2A, 2C and 2E) to quantify true positive and true negatives and F1 scores (Fig 2B, 2D and 2F) to assess false positives and false

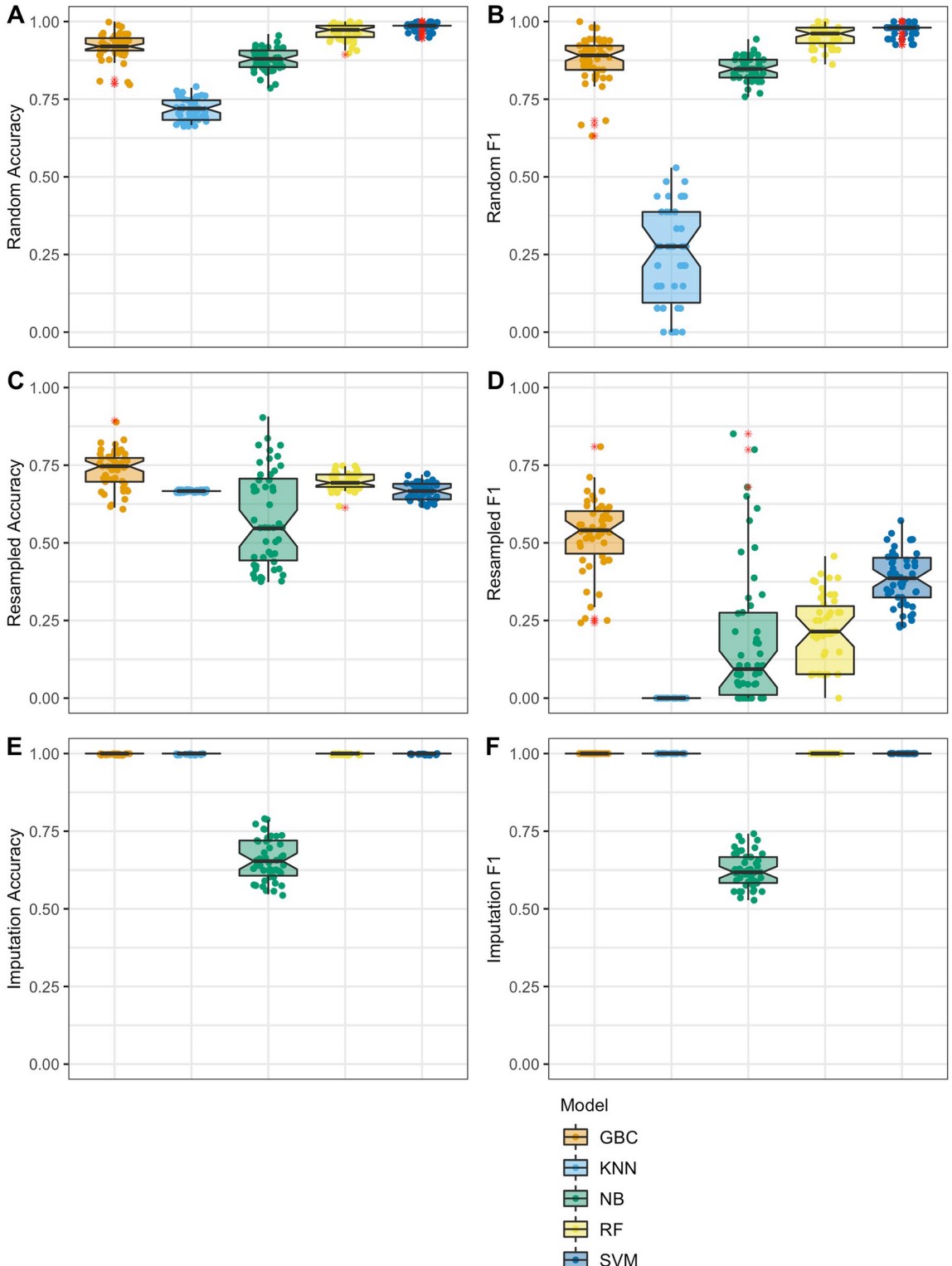

**Fig 2. Classification accuracy using copy number data.** Fabricated data was mixed with real data and given to four machine learning models for classification. Data shown represents 50 trials for 50 different fabricated dataset mixes. Features in this dataset are the copy number values for each sample. Outliers are shown as red asterisks; these same outliers are shown also as normally colored points in jittered-point overlay. **A**. Results for data fabricated with the random method, mean classification accuracy: RF 97% (+/- 2.5%), SVM 98% (+/- 1.5%), GBC 92% (+/- 4.2%), NB 88% (+/- 3.5%), KNN 72% (+/- 3.4%). **B**. Results for data fabricated with the random method,

mean classification F1: RF 0.95 (+/- 0.03), SVM 0.98 (+/- 0.02), GBC 0.88 (+/- 0.07), NB 0.85 (+/- 0.04), KNN 0.25 (+/- 0.16) **C**. Results for data fabricated with the resampling method, mean classification accuracy: RF 70% (+/- 2.6%), SVM 67% (+/- 2.7%), GBC 74% (+/- 6%), NB 58% (+/- 15.2%), KNN 67% (+/- 0%). **D**. Results for data fabricated with the resampled method, mean classification F1: RF 0.21 (+/- 0.12), SVC 0.38 (+/- 0.09), GBC 0.53 (+/- 0.12), NB 0.19 (+/- 0.23), KNN 0 (+/- 0). **E**. Results for data fabricated with the imputation method, mean classification accuracy: RF 100% (+/- 0%), SVM 100% (+/- 0%), GBC 100% (+/- 0%), NB 66% (+/- 6.7%), KNN 100% (+/- 0%). **F**. Results for data fabricated with the imputation method, mean classification F1: RF 1 (+/- 0), SVM 1 (+/- 0), GBC 1 (+/- 0), NB 0.62 (+/- 0.05), KNN 1 (+/- 0).

negatives. To ensure that our results represent robust performance, model training and evaluation was performed 50 times; each time a completely new set of 25 fabricated samples were made (see Methods). Reported results represent the average accuracy of these 50 trials.

The five models overall performed relatively well on the classification task for data fabricated with the random approach. The average accuracy scores of 50 trials was: RF 96%, SVM 98%, GBC 92%, NB 88%, and KNN 72% (Fig 2A). Mean classification accuracies were lower for data created with the resampling method, with most models losing anywhere from 5–31% accuracy (RF 70%, SVM 67%, GBC 74%, NB 58%, and KNN 67%) (Fig 2C). Since the resampling method uses data values from the real data, it is possible that fake samples very closely resemble real samples. Imputation classification accuracy results were quite high (RF100%, SVM 100%, GBC 100%, NB 66%, KNN 100%). While RF, GBC and KNN all increased in accuracy compared to the resampled data, NB performed more or less at the expected baseline accuracy (Fig 2E).

## Machine learning with digit frequencies

We were unsatisfied with the classification accuracy of the above models. One challenge for machine learning in our data is that the number of features (17,156) far exceeds the number of samples (75). In situations similar to this with high dimensionality data, feature reduction techniques can be used to reduce the number of features to increase performance and or decrease training time an example of this is principal component analysis [35]. We therefore explored ways to reduce or transform the feature set, and also to make the feature set more general and broadly applicable. Intrigued by the success of digit frequency methods in the identification of financial fraud [21], we evaluated whether this type of data representation could work for bioinformatics data as well. Therefore, all copy number data was transformed into 20 features, representing the digits 0–9 in the first and second place after the decimal of each gene copy number value. While Benford's Law describes the frequency of the first digit, genomics and proteomics data are frequently normalized or scaled and so the first digit may not be as characteristic. The shift to use the digits after the decimal point rather than the leading digit is necessary because of the constraint that Benford's law works (best) for numbers spanning several orders of magnitude. Because of the normalization present in the CNV data, the true first digits are bounded, for this reason we use the first and second digits after the decimal place, the first unbounded digits in the dataset. This is a data set specific adjustment and variations on it may need to be considered prior to its application on future datasets. For example in a dataset composed mainly of numbers between 0 and 0.09, you may need to use the third and fourth decimal point digits. Due to this adjustment, our method may be accurately referred to as Benford's Law inspired or Benford-like. These digit frequency features were tabulated for each sample to create a new data representation and fed into the exact same machine learning training and testing routine described above. Each of these 20 new features contain decimal values ranging from 0.0 to 1.0 representative of the proportional frequency that digit occurs. For example, one sample's value in the feature column for the digit 1 may

contain the value 0.3. This means that in this sample's original data the digit 1 occured in the first position after the decimal place 30% of the time.

In sharp contrast to the models built on the quantitative copy number data with random and resampled data, machine learning models which utilized the digit frequencies were highly accurate and showed less variation over the 50 trials (Fig 3). When examining the results of the data fabricated via imputation, the models achieved impressively high accuracy despite using drastically less information than those trained with the quantitative copy number values. As an average, accuracy for the 50 trials on the imputed data, RF, SVM, and the GBC models achieved 100% accuracy. The NB and KNN models were highly successful with a mean classification accuracy 98% and 96% respectively.

## Machine learning with limited data

With 17,156 CNA gene measurements, the digit frequencies represent a well sampled distribution. Theoretically, we realize that if one had an extremely limited dataset with CNA measurements for only 10 genes, the sampling of the frequencies for the 10 digits will be poor. To understand how much data is required for a good sampling of the digit-frequencies, using the imputed data, we iteratively downsampled our measurements from 17,000 to 10, (1700 was used instead of the full 17,156 since there is would be no way to do multiple unique permutations selecting 17,156 features from a set of 17,156 features). With the gene-features remaining in each downsample, the digit frequencies were re-calculated. Downsampling was performed uniformly at random without replacement. For each measurement size 50 replicates were run, all with different permutations of the downsamples. Results from this experiment can be seen in Fig 4. The number of gene-features used to calculate digit frequencies does not appear to make a difference at n > 500. In the 100 gene-feature trial, both NB and KNN have a drastic drop in performance, while the RF and GBC model remained relatively unaffected down to approximately 40 features. Surprisingly, these top performing models (GBC and RF) do not drop below 95% accuracy until they have less than 20 gene-features.

One hesitation for using machine learning with smaller datasets (i.e. fewer gene-features per sample) is the perceived susceptibility to large variation in performance. As noted, these downsampling experiments were performed 50 times, and error bars representing the standard error are shown in Fig 4. We note that even for the smallest datasets, performance does not drastically vary between the 50 trials. In fact the standard error for small datasets (e.g. 20 or 30 gene-features) is lower than when there were thousands. Thus we believe that the digit-frequency based models will perform well on both large-scale omics data and also on smaller 'targeted' data acquisition paradigms like multiplexed PCR or MRM proteomics.

## Discussion

We present here a proof of concept method for detecting fabrication in biomedical data. Just as has been previously shown in the financial sector, digit frequencies are a powerful data representation when used in combination with machine learning to predict the authenticity of data. Although the data used herein is copy number variation from a cancer cohort, we believe that the Benford-like digit frequency method can be generalized to any tabular numeric data. While multiple methods of fabrication were used, we acknowledge there are more subtle or sophisticated methods. We believe that fraud detection methods, like the models presented herein, could be refined and generalized for broad use in monitoring and oversight.

The model described here is trained to operate specifically on CNA data. However, using digit frequencies as the feature transformation creates the option to train a model on multiple data sources with different numbers of features. Here we used the copy number measurements

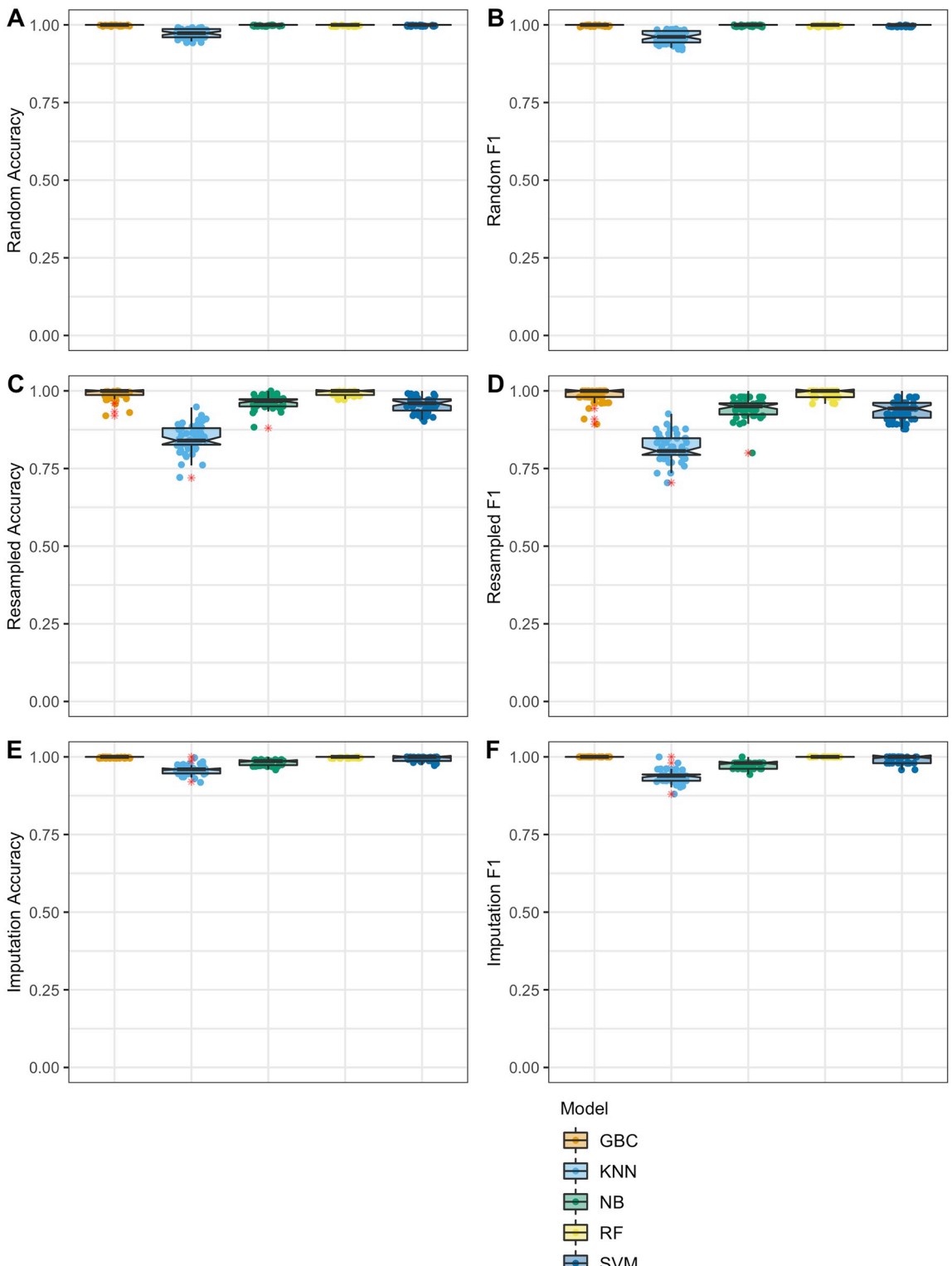

**Fig 3. Classifications accuracy using digit frequency data.** Fabricated data was mixed with real data and given to four machine learning models for classification. Data shown represents 50 trials for 50 different fabricated dataset mixes. Features in this dataset are the digit frequencies for each sample. The red asterisk represents outliers in the boxplot; these same outliers are shown as normally colored points in jittered-point overlay. **A**. Results for data fabricated with the random method, mean classification accuracy: RF 100% (+/- 0%), SVM 100% (+/- 0%), GBC 100% (+/- 0%), NB 100% (+/- 0%), KNN 97% (+/- 1.3%). **B**. Results for data fabricated with the

random method, mean classification F1: RF 1 (+/- 0), SVM 1 (+/- 0), GBC 1 (+/- 0), NB 1 (+/- 0), KNN 0.96 (+/- 0.02) **C**. Results for data fabricated with the resampling method, mean classification accuracy: RF 99% (+/- 0.8%), SVM 95% (+/- 2.3%), GBC 99% (+/- 1.7%), NB 96% (+/- 2.1%), KNN 85% (+/- 4.4%). **D**. Results for data fabricated with the resampled method, mean classification F1: RF 0.99 (+/- 0.01), SVM 0.94 (+/- 0.03), GBC 0.98 (+/- 0.02), NB 0.95 (+/- 0.03), KNN 0.82 (+/- 0.04) **E**. Results for data fabricated with the imputation method, mean classification accuracy: RF 100% (+/- 0%), SVM 100% (+/- 0.7%), GBC 100% (+/- 0%), NB 98% (+/- 0.7%), KNN 96% (+/- 1.5%). **F**. Results for data fabricated with the imputation method, mean classification F1: RF 1 (+/- 0), SVM 0.99 (+/- 0.01), GBC 1 (+/- 0), NB 0.97 (+/- 0.01), KNN 0.94 (+/- 0.02).

for 17,156 genes, but since these measurements are transformed into 20 features representing digit frequencies, theoretically, various CNA datasets with any number of measures could be used for training or testing. Just as Benford demonstrated that diverse, entirely unrelated datasets followed the same distribution of first digit, we are hopeful the same stands true for large biological datasets. However, further research would be needed to determine if a model trained on digit-frequencies of one type of omics data could be generalized and be used on another. The generalizability to such situations would likely depend on the digit distributions of the other datasets. One way to circumvent this dataset specific dependency may be to create statistical tests or use unsupervised clustering algorithms that operate within a single dataset. Moreover, future work on feature selection could potentially simplify the classification further and avoid machine learning.

A logical and exciting next step is to use this model on real published data and search for cases of fraud. There are several challenges standing in the way of doing this quickly and

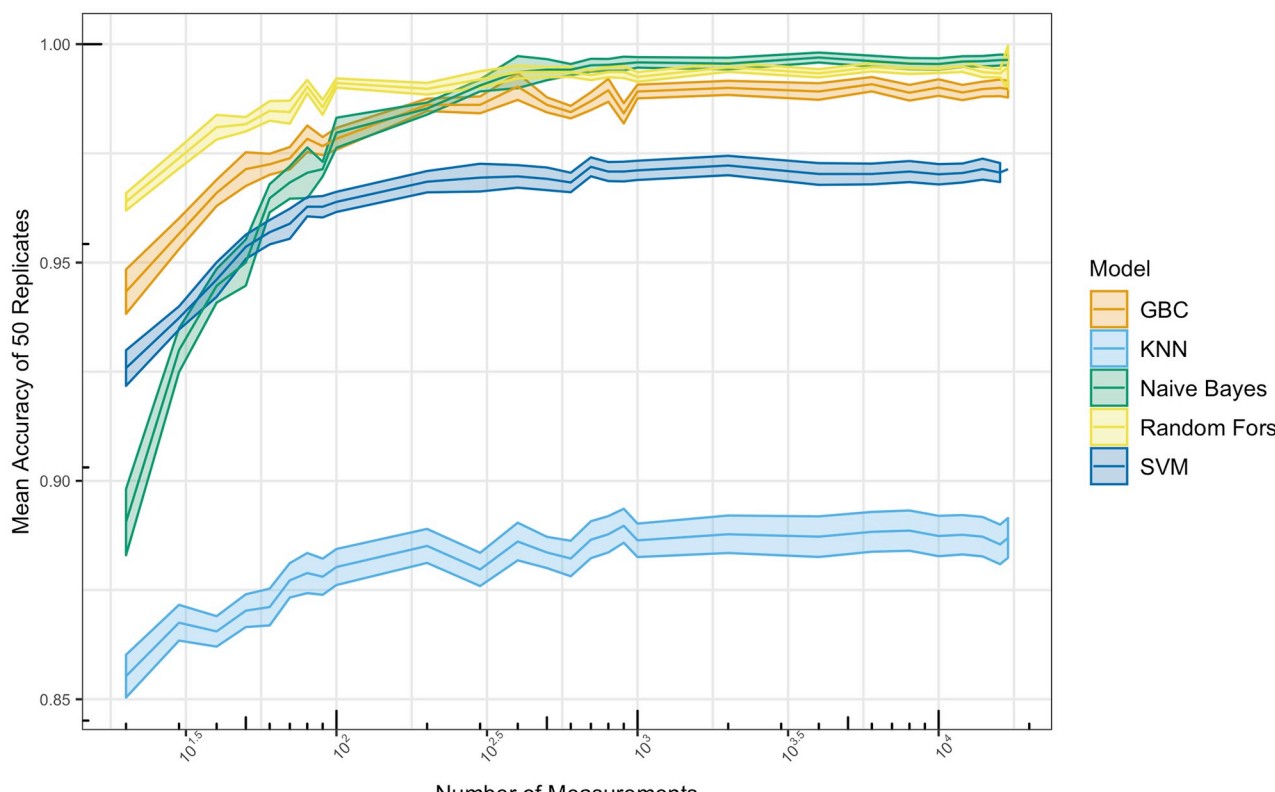

**Fig 4. Classification accuracy vs number of features.** The original 17,156 CNA measurements in the imputed dataset were randomly downsampled incrementally from 17,000 to 10 and converted to digit-frequency training and test features for machine learning models. When 1,000+ measurements are used in the creation of digit-frequencies features, there appears to be little to no effect on mean accuracy. Once the number of features drops below 300 all models begin to lose accuracy rapidly. RF remained above 97.5% accuracy until less than 30 measurements were included.

effectively. First, is the access to data. Not all journals require that data associated with a publication be made accessible and some journals that do require data accessibility count a statement to the effect of "data will be made available upon request to the authors" as sufficient—which we would argue does not constitute accessible data. Second, is the format of data. Here we used tabular CNA data generated from a large sequencing experiment, but there are numerous complex steps separating the original fastq files from nice tabular CNA data, which brings us to a third challenge. Third, reproducibility of data. Unless the study provides the tabulated form of the data or has perfectly reproducible methods for processing the rawest data, it would be difficult to know if the data being fed into the model is the exact same as that used in a study's analysis.

In order to test this method on real data, we attempted to find retracted papers known to have committed fraud. Retraction Watch (www.retractionwatch.com) maintains a large searchable database of retracted papers which aided in this task. Unfortunately, once retracted, an article and it's associated supplemental material is typically no longer available from the journal. We were able to locate some retracted papers in their original form through Sci-Hub, and within a few of these papers we were able to get URLs that were still active and pointing to where their paper's data was deposited. This however presented more challenges in the form of inconsistent formats, incomplete records (data provided for some but not all of the analyses), conversion from PDF file format to tables, and an enormous amount of manual curation.

In order for methods like this to be used broadly for data monitoring, it would require all data to be truly publicly available, in usable formats, and/or with readily reproducible methods. Even if this mass testing and monitoring of data with methods as presented was possible at this time, it should not be used as the sole determinant of trueness or falseness of a dataset; we have shown this method to be very accurate, but not perfect. The possibility of false-positives and false-negatives still exists.

A consideration in choosing to publish a method like this is the possibility it could be used for its opposite purpose and aid those attempting to commit fraud by providing a means of evaluating the quality of their data fabrication. If we had built a ready to use, easy to install and run tool, for this purpose, we would not publicly publish it. The methods we present here are a proof of concept, not a complete product. Despite being completely open source and transparent, we anticipate it would still require a great deal of time, effort, and talent to repurpose our code for something other than simply reproducing our results. We expect anyone with the required amount of time and talent could instead produce their own real data and research. To those in the future that build upon and further this type of work, we encourage you to also consider if you should publish it or not.

There is an increasing call for improved oversight and review of scientific data [5, 6, 16, 18], and various regulatory bodies or funding agencies could enforce scientific integrity through the application of these or similar methods. For example, the government bodies charged with evaluating the efficacy of new medicine could employ such techniques to screen large datasets that are submitted as evidence for the approval of new drugs. For fundamental research, publishers could mandate the submission of all data to fraud monitoring. Although journals commonly use software tools to detect plagiarism in the written text, a generalized computational tool focused on data could make data fraud detection equally simple.

## Supporting information

**S1 Fig. Methods of data fabrication.** (A) The random method of data fabrication identifies the range of observation for a specific locus and then randomly chooses a number in that range. (B) The resampling method chooses values present in the original data. (C) The

imputation method iteratively nullifies and then imputes data points from a real sample.
(TIF)

**S2 Fig. Training and testing overview.** After creating 50 fake samples using any one of the three methods of fabrication, the 100 real samples and 50 fake samples were randomly split into a train and test set of equal size and proportions (50 real and 25 fake in each set). The training sets were then used to train various machine learning models using 10-fold cross validation. Next, trained models were used to make predictions on the testing data. Predictions were then scored with total accuracy.
(TIF)

**S3 Fig. Distribution of first digits.** Distribution of normalized first-digit after the decimal frequencies in 75 real copy-number samples (A) and 50 fake samples generated by the random (B), resampled (C) and imputed (D) methods of fabrication. The x-axes represents each digit in the first position after the decimal place. The y-axes represents the normal frequency of the digit. Black lines represent the mean and diamonds represent outliers. Similar to what is seen in a distribution of first digits conforming to Benford's Law, the CNA data also exhibits a long-right tail.
(TIF)

**S4 Fig. Data relationships in fabricated data.** The correlation between pairs of genes is evaluated to determine whether fabrication methods can replicate inter-gene patterns. Plots on the left hand side (A,C,E, and G) display data from two correlated genes PLEKHN1 and HES4, adjacent genes found on 1p36. Plots on the right hand side (B,D,F, and H) display genes DFFB and OR4F5 gene with marginal Spearman correlation in the real data (.27). The plots reveal that random and resample data have little to no correlation between related genes. Imputation produces data with correlation values that are similar to the original data (.97 and.35, respectively).
(TIF)

**S1 File. Parameters for models.** Contains the hyperparameters used for all machine learning models depending on the type of data used.
(TXT)

## Author Contributions

**Conceptualization:** Samuel H. Payne.

**Formal analysis:** Michael S. Bradshaw.

**Investigation:** Michael S. Bradshaw.

**Methodology:** Michael S. Bradshaw.

**Software:** Michael S. Bradshaw.

**Supervision:** Samuel H. Payne.

**Writing – original draft:** Michael S. Bradshaw, Samuel H. Payne.

**Writing – review & editing:** Michael S. Bradshaw, Samuel H. Payne.

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
