## [Decision Letter · Decision Letter 0]

19 Feb 2021

PONE-D-20-32745

Detecting fabrication in large-scale molecular omics data

PLOS ONE

Dear Dr. Bradshaw,

Thank you for submitting your manuscript to PLOS ONE and apologies for the extended reviewing time. After careful consideration, we feel that it has merit but does not fully meet PLOS ONE’s publication criteria as it currently stands. Therefore, we invite you to submit a revised version of the manuscript that addresses the points raised during the review process.

The worth of your approach is unquestionable yet, several aspects of your manuscript lack depth and maturity, as established by both reviewers. For example, the applicability of the method and its limitations are not addressed.

Another example is the introduction of the Benford's law which raised many questions in the review process.

We look forward to receiving your revised manuscript.

Kind regards,

Frederique Lisacek

Academic Editor

PLOS ONE

Journal Requirements:

2.We note that the grant information you provided in the ‘Funding Information’ and ‘Financial Disclosure’ sections do not match.

Reviewers' comments:

Reviewer's Responses to Questions

**Comments to the Author**

1. Is the manuscript technically sound, and do the data support the conclusions?

Reviewer #1: Partly

Reviewer #2: Partly

2. Has the statistical analysis been performed appropriately and rigorously? 

Reviewer #1: Yes

Reviewer #2: No

3. Have the authors made all data underlying the findings in their manuscript fully available?

Reviewer #1: Yes

Reviewer #2: Yes

4. Is the manuscript presented in an intelligible fashion and written in standard English?

Reviewer #1: Yes

Reviewer #2: Yes

5. Review Comments to the Author

Reviewer #1: The authors present an interesting approach to detect falsification in big datasets by means of Machine Learning and a well-known feature, Benford digit preferences.

- In the "Methods" section, authors should make it explicit (or much clearer) that they are comparing TWO approaches based on ML: the first one using the actual copy-number values (not really raw data) as inputs and the second using extracted features (digit frequency)as inputs (not doing it clearly may induce the readers to confusion when reading the corresponding results section).

- In the MLTraining section, the authors mention 6 ML methods they have chosen to evaluate but they don't offer any insight on the reasons behind such a choice: Why did they selected those 6 ML methods ? Were they already known for performing well for this kind of data/problem? Were they expected to perform better than other methods? Were they selected to represent a diverse-enough palette of ML methods?.

- In addition to this, the authors should provide a short description and some literature pointers about these methods as they may allow the reader to better understand what these methods do (if not how). E.g., saying that Random Forest is an ensemble decision-tree based method, or that KNN is based on proximity/similarity in the input space and does not actually perform any "learning"... and so on for GBD, NB, MLP, and SVM.

- In the Benford-like Digit Preferences section, authors should mention that this operation is a relatively simple “feature extraction” operation (I mean, it's not trivial, nor logic, but simple). So, it is this well-informed feature extraction which allows ML models to improve their predictive performance. In addition, they should assess how far the sole use of that feature reduces the prediction problem to a simple classification problem where ML is not really necessary. Would a much simpler method produce similar results?

- In the ML with quantitative data section, the authors mention that they "evaluated the model on simple accuracy". I don't think that using simple accuracy is very informative. In the context of fake detection it should be important to assess how many false negatives and false positives are detected. I would propose the use of F1 metrics as much more informative (thus adequate) than accuracy. In addition, it should be expected that “real-world” data would have a very different distribution of fake-real samples making accuracy even less adequate (or predictive).

- Authors say that SVM and MLP performed poorly. It is a bit surprising that these 2 methods had exhibited such a poor performance, but authors don't elaborate more on this: have they tried to investigate why? Could it be due to poor configuration efforts? I feel it was too "easy" to simply exclude them from further analysis.

- What's the meaning of the red asterisks present in Figures 2 and 3? It was not possible for me to figure it out. Such kind of unexplained information may be perturbing for the readers.

- In the ML with limited data section, authors mention they downsampled data, but they don't mention for which kind of fake-generation method was that done. Although it seems it was done for resampling. Stating this is very short and would facilitate the comprehension of the experiment performed.

- The Discussion section is far too short and doesn't explore the possible implications of the proposed work nor the potential limits and reaches of the method.

Among the potential issues that should be pertinent and I would expect to be discussed are the following:

1. As mentioned before, I would expect authors to report and analyse the figures concerning the False positive and False Negatives. Even at such high accuracy values, it would be good to know if the methods under evaluation would more easily miss fake data or produce false positives.

2. In the same sense it would be interesting to determine how the methods perform when the amount of falsified data is different (either higher or lower). What happens if the distribution of falsified data in the test set is (drastically) different than the one from the training set?

Finally, authors could also (optionally) consider discussing the risk that their method could be used as "predictor" in an "adversarial attack" approach allowing to create fake data which should be detected as valid by this detector.

Reviewer #2: ### General comments

The article presents an evaluation of different machine-learning approaches to detect fraud data, and evaluates their performances on artificial fraud data generated according to three different models: random number generation, resampling and data imputation.

The article addresses an important problem for life sciences, but int its current state the evaluation suffers from several weaknesses that should be handled before publication.

In particular:

1. The three models used to generate fake data are not justified in a convincing way. Is there any reason to believe that they correspond to actual frauds? If so, examples should be provided. If not the relevance of the evaluation is questionable. Would it be possible to apply the method on actual fraud data, that has been published, detected (ans supposedly retracted)?

3. Normally, a comparative evaluation of supervised classification methods requires to tune the parameters of each of them, which was not done here. In R, you can fine tune methods for all the classical supervised classification methods (I guess similar methods exist for other language like Python) I strongly recommend to use them, identify the optimal parameters for each method, and redo the whole performance analysis. The comparison is worthless without this.

3. The main approach defended in the manuscript is to replace the actual measurements (real and fake data) by the two first decimal digits. The idea relies on Frank Benford's law, according to which the frequency distribution of leading digits from real-life sets of numerical data does not follow a uniform distribution, contrary to what might be expected.

This law is invoked like a magical trick in this context: the manuscript does not provide any explanation about the reasons for this law, it does not indicates why it would apply to the CNV data analysed here. This should be clarified. For example, it is known that one situation in which Benford's law works is for long right-tailed distributions (which is for example the case of gene expression data). The article should at least provide an histogram of the distribution of the real values and discuss its adequacy to Benford's law.

Besides, if this is the main idea, the actual distribution of leading digits should be displayed on some figure, for the real and fake data.

4. There is no indication about the usability of the method in real life conditions.

How could the ML programs be trained for real dataset? Would you recommend to generate specific fake data for each one? What about the generalization power of the approach?

What would the method give if they would be applied on a large collection of actual published data? Would some of these data set be qualified as fraud?

In summary, I think that the paper address an important issue in data science (with applications to life sciences), but in its current state it is not convincing, because of methodological weaknesses in the evaluation of performances, and because there is no indication of the relevance of the models used to generate fake data. I however think these limitations could be addressed in a revised version of the manuscript.

### Specific comments

Line 31. "When asked if their 31 colleagues had fabricated data, positive response rates rose to 14-19%"

This question is imprecise and thus the answer impossible to interpret. Does it mean that the 14-19% of the researchers personally know colleagues who fabricated data, or that they are aware of published articles where data fabrication was demonstrated (and the articles this retracted), or that they have a general awareness of the fact that data fabrication happens?

Line 57. "Frank Benford observed in a compilation of 20,000 numbers that the first digit did not follow a uniform distribution as one may anticipate"

It would be useful to explain the reason for this surprising behavior, especially since it is the basis of one of your fraud detection method.

Line 65. Section "Methods"

The computing environment should be described, in particular the language and libraries used for the analysis. I guess all this could be found on the github repository, but we have no guarantee on the long-term sustainability of a github repository, so the minimal information should be provided in the Methods section, as recommended for scientific publications.

Line 81. "Three different methods of varying sophistication are used for fabrication: random number generation, resampling with replacement and imputation"

Is there any example of actual frauds (demonstrated) that use this kind of data number generation? If yes citations should be provided. If not it question the practical relevance of the evaluation.

Line 86, section Real Data. This section should describe the dimensions of the real data set (number of features). The info comes below, but it is expected to be found here.

Has the real data been published? If so, could you provide the reference of the publication, the data repository and the accession number? Could you also provide the URL of the CPTAC portal mentioned in this section?

Line 103. "Then we iteratively nullified 10% of the data and imputed these NAs with missForrest until every value has been imputed"

What is the principle of this method? Do you impute the values based on the neighboring cells in the rows (samples), columns (features), both? This matters since the imputation should reflect the likely method used by people who generate fraud data. Moreover, the way the imputation is done is likely to affect the machine-learning performances.

L122, section "Machine learning training". It would be good to compute the performance

L96, "For every gene locus, we first find the maximum and minimum values observed in the original data. A new sample is then fabricated by randomly picking a value within this gene specific range"

and further L158. "the random data clusters far from the real data"

Do you mean you used a uniform distribution to generate random numbers? If so it is not surprising that these fake samples clusterize far away from the real data and other fake data. Why did you use such a model rather than some random number model closer to the data ? For example a multivariate normal model whose parameters (correlation matrix- have been estimated on the real data. This would be a much more relevant way to generate more relevant random numbers.

L182. The abbreviations are missing for several methods (NB, RF), whereas they are used in the text and figures.

L196. The theoretical baseline accuracy is 66% according to the training/testing class sizes. It would be worth checking empirically the untrained performances of the different ML methods, by computing the accuracy in an "untrained" mode, i.e. by randomly permuting the training and testing labels. In principle this should return accuracies of ~66%, but there are sometimes tricky issues, so it is worth testing it.

It would also be useful to plot the baseline + untrained performances on the accuracy box plots.

L182. The parameters used for each ML method should be provided (either here or in the Material and Methods section).

L190. The accuracy is not a sufficient parameter to evaluate the performances of a 2-group classifier aiming at detecting one particular case (declare as "positives" the fake samples). For each method, you should compute he sensitivity and false predictive rate. The results of the different methods could be displayed on a classical Sn / FPR plot (in addition, f you tune some quantitative parameters you could draw a ROC curve).

L 194. "SVM and MLP performed poorly compared to other classification methods". I suspect this comes from the fact that you let all the methods run with their default parameters. In particular, SVM results vary hugely depending on the choice of the kernel, and the optimal kernel is a case-by-case affair, so you should absolutely test the performance of the different kernels (linear, radial, polynomial, sigmoid).

Actually, a comparative evaluation requires to tune the parameters of each ML method, which was not done here. In R, you can fine tune methods for all the classical supervised classification methods. I strongly recommend to use them and redo the whole performance analysis.

L225. "One challenge for machine learning in our data is that the number of features (~17,000) far exceeds the number of samples (75). We therefore explored ways to reduce or transform the feature set, and also to

228 make the feature set more general and broadly applicable."

This is a very strange motivation for using a digit preference approach, which looks a bit like a magical trick in this context.

If the goal is to reduce the over-dimensionality of the feature space, a first and obvious option would have been to train the classifiers on the first components (this is a very classical approach). Another possibility would be to test any classical method for feature selection.

L225. Why are there 17.000 features in the original dataset ? There are ~50,000 genes in the current annotations of Human genome.

L232 "the decimal of each gene expression value".

Are we speaking of CNV or transcriptome ?

L245. "Converting all measured variables to digit frequencies circumvents this problem. For instance, if you had a data set of CNA and transcriptomic data a machine learning model could not train and test on both of these. "

I don't see any reason for this claim. If both CNV and expression data are real numbers (which is the case) they can perfectly be combined in a feature matrix to feed the ML methods. The fact that their range and distributions would differ might pose a problem for some methods, but most if not all of the methods you used are equipped to handle variables with different data ranges. And in any case you did not check if the data distribution would or not fit the assumptions underlying the different methods.

"The features in these datasets would differ in the number of features and what these features represent. "

This makes no sense. If you combine NCV and expression data for the same samples, the number of features (genes) should in principle be the same for the two datasets, and they should thus be balanced (and in any case, which would not even be a prerequisite for combining them). In addition, all the ML methods are classically used to analyse features representing different things (e.g. size, weight, fat content, protein content,...), this is the essence of multivariate analysis.

L254. "over the 50 trails" Did you mean "trials" ?

L283. "Surprisingly, these top performing models (GBC and Random Forest) do not drop below 95% accuracy until they have less than 20 gene-features."

Why is this surprising? This simply reflects the fact that the fake data is simple to detect, which may come from the way they are generated.

What I find surprising here is that you can learn something from the distribution of the two first digits (i.e. 10 x 10 numbers) computed from only 20 features. This means that each pair of digit is expected to be found 0.2 times in the data. I an thus very skeptical about this result, and I suspect there is a trick somewhere. I would suggest you to check how the distribution of digits evolves (separately for the real and for each fake data) as you reduce the dimension of the feature space, and to see if there is not some bias.

L281. "In the 100 gene-feature trial, both Naive Bayes and KNN have a significant drop in performance"

This drop of performances on Figure 4 may be a visual artifact resulting from the arbitrary numbers of features chosen for your analysis : you increase the number of features by steps of 10 until 100, then you jump from 100 to 500, then to 1000, 2000 and you increase by steps of 2000. If the goal is to display the impact of N on both e small and large range, you should better use an XY plot with a logarithmic X axis. Also, it would be worth exploring the region between 100 and 500, since this is the place where you claim to observe a drop. I would recommend to add a measurement of the performances with 200, 300, 400 features, respectively.

6. PLOS authors have the option to publish the peer review history of their article (what does this mean?). If published, this will include your full peer review and any attached files.

Reviewer #1: No

Reviewer #2: **Yes: **Jacques van Helden (ORCID 0000-0002-8799-8584)

---

## [Author Response · Author response to Decision Letter 0]

30 Apr 2021

A copy of this information was also uploaded as a Word Doc with our responses highlighted for ease of understanding

Reviewer #1: The authors present an interesting approach to detect falsification in big datasets by means of Machine Learning and a well-known feature, Benford digit preferences.

1 - In the "Methods" section, authors should make it explicit (or much clearer) that they are comparing TWO approaches based on ML: the first one using the actual copy-number values (not really raw data) as inputs and the second using extracted features (digit frequency)as inputs (not doing it clearly may induce the readers to confusion when reading the corresponding results section).

Response: Added to end of machine learning methods

2 - In the MLTraining section, the authors mention 6 ML methods they have chosen to evaluate but they don't offer any insight on the reasons behind such a choice: Why did they selected those 6 ML methods ? Were they already known for performing well for this kind of data/problem? Were they expected to perform better than other methods? Were they selected to represent a diverse-enough palette of ML methods?.

Response: An explanation has been added to the machine learning section explaining why these 6 were chosen

3 - In addition to this, the authors should provide a short description and some literature pointers about these methods as they may allow the reader to better understand what these methods do (if not how). E.g., saying that Random Forest is an ensemble decision-tree based method, or that KNN is based on proximity/similarity in the input space and does not actually perform any "learning"... and so on for GBD, NB, MLP, and SVM.

Response: I have added a one sentence description of each algorithm and a reference so readers can learn more if they choose.

4 - In the Benford-like Digit Preferences section, authors should mention that this operation is a relatively simple “feature extraction” operation (I mean, it's not trivial, nor logic, but simple). So, it is this well-informed feature extraction which allows ML models to improve their predictive performance. In addition, they should assess how far the sole use of that feature reduces the prediction problem to a simple classification problem where ML is not really necessary. Would a much simpler method produce similar results?

Response: It is possible that other methods of data/feature extraction could produce a similar result. However, the exploration of alternative methods is outside of the scope of our manuscript.

5 - In the ML with quantitative data section, the authors mention that they "evaluated the model on simple accuracy". I don't think that using simple accuracy is very informative. In the context of fake detection it should be important to assess how many false negatives and false positives are detected. I would propose the use of F1 metrics as much more informative (thus adequate) than accuracy. In addition, it should be expected that “real-world” data would have a very different distribution of fake-real samples making accuracy even less adequate (or predictive).

Response: Figure 2 and 3 have been updated to include accuracy and F1 scores. As a side note, the F1 score plots we generated appear extremely similar to the accuracy ones, but if you look closely they are not. All the F1 scores in the digit-preference section are very close to 1.0. This was a good recommendation, thanks.

6 - Authors say that SVM and MLP performed poorly. It is a bit surprising that these 2 methods had exhibited such a poor performance, but authors don't elaborate more on this: have they tried to investigate why? Could it be due to poor configuration efforts? I feel it was too "easy" to simply exclude them from further analysis.

Response: In the initial study hyperparameter optimization was not performed. We have added grid search parameter optimization for each model and the “optimal” sets were used for all final test set results. All results have been updated to reflect these changes. SVM was also optimized, has included in all analysis and performs comparably to all other models. The optimization of the MLP is a near infinite search space and since we have five other models that work extremely well we do not see the need to spend the time and resources optimizing the MLP and have removed MLP from the paper entirely now.

7 - What's the meaning of the red asterisks present in Figures 2 and 3? It was not possible for me to figure it out. Such kind of unexplained information may be perturbing for the readers.

Response: The red asterisk represents outliers in the boxplot. An explanation of this had been added to captions for figures 2 and 3.

8 - In the ML with limited data section, authors mention they downsampled data, but they don't mention for which kind of fake-generation method was that done. Although it seems it was done for resampling. Stating this is very short and would facilitate the comprehension of the experiment performed.

Response: It was done with imputed data, a sentence has been added explaining this

9 - The Discussion section is far too short and doesn't explore the possible implications of the proposed work nor the potential limits and reaches of the method.

Among the potential issues that should be pertinent and I would expect to be discussed are the following:

10- 1. As mentioned before, I would expect authors to report and analyse the figures concerning the False positive and False Negatives. Even at such high accuracy values, it would be good to know if the methods under evaluation would more easily miss fake data or produce false positives.

Response: This has been addressed, see previous comments

11 - 2. In the same sense it would be interesting to determine how the methods perform when the amount of falsified data is different (either higher or lower). What happens if the distribution of falsified data in the test set is (drastically) different than the one from the training set?

Response: This has been addressed in the discussion section

12 - Finally, authors could also (optionally) consider discussing the risk that their method could be used as "predictor" in an "adversarial attack" approach allowing to create fake data which should be detected as valid by this detector.

Response: we have added this to the discussion.

Reviewer #2: ### General comments

13 - The article presents an evaluation of different machine-learning approaches to detect fraud data, and evaluates their performances on artificial fraud data generated according to three different models: random number generation, resampling and data imputation.

The article addresses an important problem for life sciences, but int its current state the evaluation suffers from several weaknesses that should be handled before publication.

In particular:

14 - 1. The three models used to generate fake data are not justified in a convincing way. Is there any reason to believe that they correspond to actual frauds? If so, examples should be provided. If not the relevance of the evaluation is questionable. Would it be possible to apply the method on actual fraud data, that has been published, detected (ans supposedly retracted)?

Response: Finding an actual fraud dataset would be ideal. Prior to submitting this article I spent a good deal of time trying to find datasets from papers listed in RetractionWatch. Unfortunately, since these articles have been taken down already, finding links to where their data was deposited is difficult or impossible. 

15 - 3. Normally, a comparative evaluation of supervised classification methods requires to tune the parameters of each of them, which was not done here. In R, you can fine tune methods for all the classical supervised classification methods (I guess similar methods exist for other language like Python) I strongly recommend to use them, identify the optimal parameters for each method, and redo the whole performance analysis. The comparison is worthless without this.

Response: We added parameter optimization for each model using GridSearch from sklearn in python. All results have been rerun and reported with their optimized models and we have added additional detail to the methods section reflecting this.

16 - 3. The main approach defended in the manuscript is to replace the actual measurements (real and fake data) by the two first decimal digits. The idea relies on Frank Benford's law, according to which the frequency distribution of leading digits from real-life sets of numerical data does not follow a uniform distribution, contrary to what might be expected.

This law is invoked like a magical trick in this context: the manuscript does not provide any explanation about the reasons for this law, it does not indicates why it would apply to the CNV data analysed here. This should be clarified. For example, it is known that one situation in which Benford's law works is for long right-tailed distributions (which is for example the case of gene expression data). The article should at least provide an histogram of the distribution of the real values and discuss its adequacy to Benford's law.

Besides, if this is the main idea, the actual distribution of leading digits should be displayed on some figure, for the real and fake data.

Response: A supplemental figure of the distribution of digit frequencies in the real and fake data has been added, along with an explanation of how the CNA data is similar to a distribution following Benford’s law.

17 - 4. There is no indication about the usability of the method in real life conditions.

How could the ML programs be trained for real dataset? Would you recommend to generate specific fake data for each one? What about the generalization power of the approach?

Response: We have addressed this in new paragraphs added to the discussion section

What would the method give if they would be applied on a large collection of actual published data? Would some of these data set be qualified as fraud?

Response: This is now addressed in the discussion

In summary, I think that the paper address an important issue in data science (with applications to life sciences), but in its current state it is not convincing, because of methodological weaknesses in the evaluation of performances, and because there is no indication of the relevance of the models used to generate fake data. I however think these limitations could be addressed in a revised version of the manuscript.

### Specific comments

18 - Line 31. "When asked if their 31 colleagues had fabricated data, positive response rates rose to 14-19%"

This question is imprecise and thus the answer impossible to interpret. Does it mean that the 14-19% of the researchers personally know colleagues who fabricated data, or that they are aware of published articles where data fabrication was demonstrated (and the articles this retracted), or that they have a general awareness of the fact that data fabrication happens?

Response: I have added clarifying detail, the question in the survey was geared towards personally knowing of colleague that fabricated data

19 - Line 57. "Frank Benford observed in a compilation of 20,000 numbers that the first digit did not follow a uniform distribution as one may anticipate"

It would be useful to explain the reason for this surprising behavior, especially since it is the basis of one of your fraud detection method.

Response: an explanation has been added to the introduction

20 - Line 65. Section "Methods"

The computing environment should be described, in particular the language and libraries used for the analysis. I guess all this could be found on the github repository, but we have no guarantee on the long-term sustainability of a github repository, so the minimal information should be provided in the Methods section, as recommended for scientific publications.

Response: we have created a computing environment section of the paper and added a supplemental file with the parameters used for each ML model.

21 - Line 81. "Three different methods of varying sophistication are used for fabrication: random number generation, resampling with replacement and imputation"

Is there any example of actual frauds (demonstrated) that use this kind of data number generation? If yes citations should be provided. If not it question the practical relevance of the evaluation.

Response: Finding an actual fraud dataset would be ideal. Prior to submitting this article we spent a good deal of time trying to find datasets from papers listed in RetractionWatch. Unfortunately, since these articles have been taken down already, finding links to where their data was deposited is difficult or impossible. We have added a discussion of this and the challenges to the manuscript.

22 - Line 86, section Real Data. This section should describe the dimensions of the real data set (number of features). The info comes below, but it is expected to be found here.

Has the real data been published? If so, could you provide the reference of the publication, the data repository and the accession number? Could you also provide the URL of the CPTAC portal mentioned in this section?

Response: Citations for CPTAC have been added. All data used in our analyses also exists in our github repo. 

23 - Line 103. "Then we iteratively nullified 10% of the data and imputed these NAs with missForrest until every value has been imputed"

What is the principle of this method? Do you impute the values based on the neighboring cells in the rows (samples), columns (features), both? This matters since the imputation should reflect the likely method used by people who generate fraud data. Moreover, the way the imputation is done is likely to affect the machine-learning performances.

Response: the imputation is done based on neighboring samples (rows). The order of features (columns) is not necessarily meaningful. Additional information on how the imputations was done can be found in the paper describing the tool we use: missForest R package. Link: https://doi.org/10.1093/bioinformatics/btr597

23 - L122, section "Machine learning training". It would be good to compute the performance

L96, "For every gene locus, we first find the maximum and minimum values observed in the original data. A new sample is then fabricated by randomly picking a value within this gene specific range"

and further L158. "the random data clusters far from the real data"

Do you mean you used a uniform distribution to generate random numbers? If so it is not surprising that these fake samples clusterize far away from the real data and other fake data. Why did you use such a model rather than some random number model closer to the data ? For example a multivariate normal model whose parameters (correlation matrix- have been estimated on the real data. This would be a much more relevant way to generate more relevant random numbers.

Response: Yes, this is a simple method and we intended it as such. Our goal with the three methods of fake data generation we selected was to create fake data with varying degrees of sophistication. Our random number generation was intended to be the easiest method to detect. The changes you propose here sound similar from our third and most sophisticated method functions - imputation.

24 - L182. The abbreviations are missing for several methods (NB, RF), whereas they are used in the text and figures.

Response: abbreviations have been added where the terms are first used

25 - L196. The theoretical baseline accuracy is 66% according to the training/testing class sizes. It would be worth checking empirically the untrained performances of the different ML methods, by computing the accuracy in an "untrained" mode, i.e. by randomly permuting the training and testing labels. In principle this should return accuracies of ~66%, but there are sometimes tricky issues, so it is worth testing it.

It would also be useful to plot the baseline + untrained performances on the accuracy box plots.

26 - L182. The parameters used for each ML method should be provided (either here or in the Material and Methods section).

Response: The full list of parameters for all the models is long. Rather than include it in the text we have prepared a supplementary file (supplementary_file_1.txt) that will accompany the manuscript.

27 - L190. The accuracy is not a sufficient parameter to evaluate the performances of a 2-group classifier aiming at detecting one particular case (declare as "positives" the fake samples). For each method, you should compute he sensitivity and false predictive rate. The results of the different methods could be displayed on a classical Sn / FPR plot (in addition, f you tune some quantitative parameters you could draw a ROC curve).

Response: Reviews 1 and 2 both pointed out this weakness of our analysis but proposed slightly different figure additions to address it. Per recommendation of review 1 we have included plots of F1 scores adjacent to the accuracy plots. This addresses the same concern of accuracy alone not measuring or reporting false positives or false negatives. As a side note, the F1 score plots we generated appear extremely similar to the accuracy ones, but if you look closely they are not. All the F1 scores in the digit-preference section are very close to 1.0. 

28 - L 194. "SVM and MLP performed poorly compared to other classification methods". I suspect this comes from the fact that you let all the methods run with their default parameters. In particular, SVM results vary hugely depending on the choice of the kernel, and the optimal kernel is a case-by-case affair, so you should absolutely test the performance of the different kernels (linear, radial, polynomial, sigmoid).

Actually, a comparative evaluation requires to tune the parameters of each ML method, which was not done here. In R, you can fine tune methods for all the classical supervised classification methods. I strongly recommend to use them and redo the whole performance analysis.

Response: We added parameter optimization and all performance evaluations have been re run. Once optimized SVM performed similarly to the other models. The optimization of the MLP is a near infinite search space and since we have five other models that work extremely well we do not see the need to spend the time and resources optimizing the MLP and have removed from the paper entirely now.

29 - L225. "One challenge for machine learning in our data is that the number of features (~17,000) far exceeds the number of samples (75). We therefore explored ways to reduce or transform the feature set, and also to

228 make the feature set more general and broadly applicable."

This is a very strange motivation for using a digit preference approach, which looks a bit like a magical trick in this context.

If the goal is to reduce the over-dimensionality of the feature space, a first and obvious option would have been to train the classifiers on the first components (this is a very classical approach). Another possibility would be to test any classical method for feature selection.

Response: Feature reduction techniques are not uncommon in ML. For example: principal component analysis is used to reduce high dimensionality data. An explanation of this and an additional citation have been added.

30 - L225. Why are there 17.000 features in the original dataset ? There are ~50,000 genes in the current annotations of Human genome.

Response: this is the data from the CPTAC dataset. See their publications for a further explanation [PMID: 33560848, PMID: 32059776]

31 - L232 "the decimal of each gene expression value".

Are we speaking of CNV or transcriptome ?

Response: Agreed, this statement was unclear / incorrect. Updated to “gene copy number value”

32 - L245. "Converting all measured variables to digit frequencies circumvents this problem. For instance, if you had a data set of CNA and transcriptomic data a machine learning model could not train and test on both of these. "

I don't see any reason for this claim. If both CNV and expression data are real numbers (which is the case) they can perfectly be combined in a feature matrix to feed the ML methods. The fact that their range and distributions would differ might pose a problem for some methods, but most if not all of the methods you used are equipped to handle variables with different data ranges. And in any case you did not check if the data distribution would or not fit the assumptions underlying the different methods.

"The features in these datasets would differ in the number of features and what these features represent. "

This makes no sense. If you combine NCV and expression data for the same samples, the number of features (genes) should in principle be the same for the two datasets, and they should thus be balanced (and in any case, which would not even be a prerequisite for combining them). In addition, all the ML methods are classically used to analyse features representing different things (e.g. size, weight, fat content, protein content,...), this is the essence of multivariate analysis.

Response: If the number of genes (features) in two datasets are not the same (as would likely happen if they were from different experiments) you could not train on one and then the next because the input data shapes would not match. For example we have ~17,000 features in our CNA data, if we trained a model initially on this dataset we could not then test or train on a data set with 10,000 features; the models cannot handle this. However if you transform and reduce the ~17,000 into the 20 features of digit-preferences and then did the same for the 10,000 you could train on one and test or train more with the same model. This is just a question of the shape of the input data. 

There is already an explanation of this in our manuscript: “Thus for each sample the features are converted from 17,156 copy number alterations to 20 digit preferences. Using this approach, whether a sample has 100 or 17,156 features it can still be trained on and classified by the same model."

33 - L254. "over the 50 trails" Did you mean "trials" ?

Response: Yes, fixed.

34 - L283. "Surprisingly, these top performing models (GBC and Random Forest) do not drop below 95% accuracy until they have less than 20 gene-features."

Why is this surprising? This simply reflects the fact that the fake data is simple to detect, which may come from the way they are generated.

What I find surprising here is that you can learn something from the distribution of the two first digits (i.e. 10 x 10 numbers) computed from only 20 features. This means that each pair of digit is expected to be found 0.2 times in the data. I an thus very skeptical about this result, and I suspect there is a trick somewhere. I would suggest you to check how the distribution of digits evolves (separately for the real and for each fake data) as you reduce the dimension of the feature space, and to see if there is not some bias.

Response: We understand the reviewers hesitancy at our result. We were also impressed at the performance. But we note that, as cited in our introduction, simple digit frequencies have been very successful at finding fraud in financial and other numeric data. Additionally, we note our complete transparency in the analysis and reporting. All of the data has been open since the first submission to bioRxiv almost 2 years ago. All figures are made with publicly available code, and can be manually inspected or verified.

35 - L281. "In the 100 gene-feature trial, both Naive Bayes and KNN have a significant drop in performance"

This drop of performances on Figure 4 may be a visual artifact resulting from the arbitrary numbers of features chosen for your analysis : you increase the number of features by steps of 10 until 100, then you jump from 100 to 500, then to 1000, 2000 and you increase by steps of 2000. If the goal is to display the impact of N on both e small and large range, you should better use an XY plot with a logarithmic X axis. Also, it would be worth exploring the region between 100 and 500, since this is the place where you claim to observe a drop. I would recommend to add a measurement of the performances with 200, 300, 400 features, respectively.

Response: we have increased the granularity of this figure to include all 100 feature steps from 100-1000 features and used a log scale x axis. 

6. PLOS authors have the option to publish the peer review history of their article (what does this mean?). If published, this will include your full peer review and any attached files.

Do you want your identity to be public for this peer review? For information about this choice, including consent withdrawal, please see our Privacy Policy.

Reviewer #1: No

Reviewer #2: Yes: Jacques van Helden (ORCID 0000-0002-8799-8584)

---

## [Decision Letter · Decision Letter 1]

17 Sep 2021

PONE-D-20-32745R1

Detecting fabrication in large-scale molecular omics data

PLOS ONE

Dear Dr. Bradshaw,

Thank you for submitting your manuscript to PLOS ONE and for your patience. This manuscript is very well received and the stakes are rather high if such work is not given the attention it deserves. The selection of fair and expert reviewers is the main reason for the delay. This expertise is rare.

After careful consideration, we feel that it has merit but does not fully meet PLOS ONE’s publication criteria as it currently stands. Therefore, we invite you to submit a revised version of the manuscript that addresses the points raised during the review process.

The first round of reviews revealed issues that you have attended to and the new reviewer spotted a last issue regarding the processing of real datasets that you need to consider. This is rather minor in terms of effort on your part and will be major in terms of impact.

We look forward to receiving your revised manuscript.

Kind regards,

Frederique Lisacek

Academic Editor

PLOS ONE

Journal Requirements:

Reviewers' comments:

Reviewer's Responses to Questions

**Comments to the Author**

1. If the authors have adequately addressed your comments raised in a previous round of review and you feel that this manuscript is now acceptable for publication, you may indicate that here to bypass the “Comments to the Author” section, enter your conflict of interest statement in the “Confidential to Editor” section, and submit your "Accept" recommendation.

Reviewer #1: All comments have been addressed

Reviewer #3: (No Response)

2. Is the manuscript technically sound, and do the data support the conclusions?

Reviewer #1: Yes

Reviewer #3: Partly

3. Has the statistical analysis been performed appropriately and rigorously? 

Reviewer #1: Yes

Reviewer #3: Yes

4. Have the authors made all data underlying the findings in their manuscript fully available?

Reviewer #1: Yes

Reviewer #3: Yes

5. Is the manuscript presented in an intelligible fashion and written in standard English?

Reviewer #1: Yes

Reviewer #3: Yes

6. Review Comments to the Author

Reviewer #1: 2nd review

----------

- Response 4: "the exploration of alternative methods is outside of the scope of our manuscript"  I was expecting more a comment about that possibility than a deep exploration. One could discuss such issue based for example on the fact that k-NN (the less ML of the methods presented) also improves drastically (significantly :-) its predictive performance.

- A comment I didn't think of on the first review that could enrich the discussion concerns the potential of using a similar approach, but in an unsupervised (or semi-supervised) manner for detecting "anomalies" in datasets so as to flag potential falsifications (as done in fraud or cyberattack contexts) without having a training set of already known fake-data strategies. (For future work, not to be addressed now)

- Finally, I haven't addressed the responses to the comments from the 2nd reviewer as I find he or she would be the best placed to judge on their quality.

Minor comments

- Line 131: per-se instead of per-say

- Line 233: we tested FIVE different ,methods (not six)

- Line 247: "the remaining four models"  Not clear which are the "remaining" models or even why is that word used here.

Reviewer #3: I love the idea behind this paper. Fraud is a very significant problem within scientific research, particularly for increasingly data right subject areas, and should be a concern for all of us in this community. Taking steps to develop tools to detect fraud is a key pillar of addressing this issue, alongside good Open Science practices that ensure transparency and replicability throughout the research chain (including in peer review!). Unfortunately, however, I have some significant concerns about this manuscript as it stand and I'm not convinced it's ready to be published as it stands. I outline those concerns below. First though, I would like to strongly encourage the authors to continue developing this manuscript, despite the significant review and publication delays since the first BioRxiv preprint. I have no doubt this will be a valuable piece of work in this important area.

Primary concerns:

1. The authors use three different mechanisms for generating fake data, random number generation, resampling and imputation. These are implemented with the aim of approximating the real data as accurately as possible. It is far from clear to me that any of these strategies reflect strategies scientists would actually use to fake data, however they are as reasonable as any other strategies given the lack of evidence base on this. The way they are used however is where my concern lies.

What motivation would someone have to fake data that reflects the real data and doesn't generate any clear 'result'?!? What the authors have here is a model that detects simulated data with the same characteristics as the current data. I suppose it is possible that scientists might want to fake (simulate) extra samples with similar statistical characteristics as their real data so as to inflate the sample number in their experiments, but it seems far more likely to me that scientists would try to fake data to generate a result.

For this paper, the simplest fake result to add to the data would be a shift in the CNV value to higher or lower values for specific subsets of samples or (more likely) for specific genes within specific subsets of sample. This more realistic test would be simple to implement within the three methods used here. To summarise; in order to be convinced that these methods are useful, I want to see their performance on a real world dataset with a CNV-treatment result in it, with different types of faked data (global up-/down- & specific gene up-down) added to either enhance/deplete the significance of the result, or to add new results to the data. I'd also like to see how models trained on the fake data with these signals in perform; retraining of the models here would probably require a more nuanced investigation for the training in order to avoid training the models just to recognise the up-/down-regulation, rather than the other characteristics of the fake data.

2. For the model trained on the two decimal digits, the models are essentially being trained to detect data that doesn't obey Benford's-law. The authors haven’t demonstrated that the machine learning models outperform the far simpler process of making the appropriate histogram and fitting a curve based on Benford's law to this and seeing if you get a decent fit (with a KS test, for example). It's possible that the ML models outperform this simple test, but the authors need to do this comparison to motivate the use of the more complex and opaque ML algorithms.

3. Figures 2 & 3 contain boxplots suggesting that some of the models have zero variation in their performance across different data subsets. In some cases this is because the clarifiers are apparently perfectly good/bad accuracy (which I am deeply suspicious of and seems too good to be true) but in some cases it's perfectly consistent accuracy performance (e.g. Fig 2 panel C). These results seem to be in disagreement with Figure 4 which suggests that the average accuracy performance never reaches 100% for any of the models. Something isn't right here. The authors need to carefully inspect their methods reconcile these figures, and either convincingly justify the perfectly good/bad/consistent performance or (more likely) fix the bug that’s causing these.

Detail comments:

1. Line 69/70. I think the readers would benefit from adding some clarity on the limitations of Benford's law here. In particular, it's only really valid for data that spans several orders of magnitude, and for data where the upper/lower limits are not tightly bounded.

2. Line 86. I disagree with the statement that "making up data is always wrong"; a bit more nuance is needed here. Firstly, simulating data has a long history of being informative in many areas of science. Secondly, there is a grey area here around imputed data, and particularly the imputation of missing data. It is, for example, commonplace to model a covariate in order to impute missing values in this data, and then to use this covariate data - including the imputed data - in a second model which leads to interpretation. From a certain perspective (my perspective, for example!) this could be seen as 'making up data' that has a direct impact on results/conclusions (depending on the scale of the missing data). This is certainly not widely considered wrong or inappropriate.

3. Line 111-113: does this mean that some of the samples are represented twice in the 150 sample dataset, albeit with 10% imputed data? How do we know that the ML models are learning to separate the fake samples from the real based on the imputed data signal, rather than needing both a duplicated sample and the imputed data signal. If you added samples that aren't in the original data, with a 10% imputation, would the performance of the models be as good?

4. Line 166: "Machine learning cannot…" I know what you're getting at, but this is not well worded. I think you wat to say something like : "Trained ML models are restricted to data that conform to the model input specifications (i.e. the same number of input features, for example).

5. Line 168: I think it would be worth noting here that the generalizability of this model comes with a cost - it will only work for data where Benfords-law should be valid, which is certainly not all datasets.

6. Line 177: "tiddy-verse" should be "tidyverse"

7. Line 233: "six" this should be five - this needs checking throughout the paper since the number of models used has changed through the review process.

8. Line 247: "The remaining four models…". I think this region of text has been re-ordered quite a bit during the review process and it doesn't make much sense now since we haven't has the results for the fifth model yet at this point. I think the authors need to give this section a careful read and make sure it flows sensibly.

9. Line 286: "[29149684]" I think this should be a reference??

10. Line 289: "While Benford's law…" The shift to use the decimal point digits rather than the leading digit is necessary because of the constraint that Benford law works (best) for numbers spanning several orders of magnitude. This is not the case for the first digit in the CNV data, but this value is usually a non-zero value so the first and second digits necessarily span orders of magnitude. This is a dataset specific approach though. in a dataset comprised mainly of numbers between 0 and 0.09, you would need to use the third and fourth decimal point digits. This would be work illuminating here.

11. Line 301-307: "Machine learning typically…" repetition of previous text and discussion. I think this can be removed.

12. Line 339. I'm not very surprised at the reasonable performance with as few as 10 genes here. 10 genes x 75 samples = 750 datapoints. This is plenty to build a histogram to compare with the Benfords Law curve (the equivalent of which is what the ML models are learning to do) .

13. Line 353: I think "per data point" should be "per sample" here.

14. Figure 2: I can't really see the details of this figure well - the resolution is quite low in the PDF embedding. It would be useful to explain the components of the box plot (median, quartiles, indents, etc) for those not familiar with boxplots.

15. Supp. Fig. 3. This figure is really useful (I'd put it in the main paper) but it's a nightmare to read because it's very busy. I suggest that the authors split the figure into four facet panels with one dataset per panel.

7. PLOS authors have the option to publish the peer review history of their article (what does this mean?). If published, this will include your full peer review and any attached files.

Reviewer #1: **Yes: **Carlos Peña-Reyes

Reviewer #3: **Yes: **Dr Nicholas Schurch

---

## [Author Response · Author response to Decision Letter 1]

8 Nov 2021

Response to Reviews

Below is a detailed response to reviewer critiques.

- Bradshaw and Payne

Reviewer #1: 2nd review

----------

Reviewer #1 has asked for an extended discussion of two points. We have addressed each of these within the manuscript. 

- Response 4: "the exploration of alternative methods is outside of the scope of our manuscript"  I was expecting more a comment about that possibility than a deep exploration. One could discuss such issue based for example on the fact that k-NN (the less ML of the methods presented) also improves drastically (significantly :-) its predictive performance.

We have inserted some text that discusses this point.

- A comment I didn't think of on the first review that could enrich the discussion concerns the potential of using a similar approach, but in an unsupervised (or semi-supervised) manner for detecting "anomalies" in datasets so as to flag potential falsifications (as done in fraud or cyberattack contexts) without having a training set of already known fake-data strategies. (For future work, not to be addressed now)

We are grateful for this suggestion and have added a bit about the possibility of cluster and a couple citations to work on it in other fields. 

Minor comments

We have fixed all the minor typographical errors noted

Reviewer #3: 

I love the idea behind this paper. Fraud is a very significant problem within scientific research, particularly for increasingly data right subject areas, and should be a concern for all of us in this community. Taking steps to develop tools to detect fraud is a key pillar of addressing this issue, alongside good Open Science practices that ensure transparency and replicability throughout the research chain (including in peer review!). Unfortunately, however, I have some significant concerns about this manuscript as it stand and I'm not convinced it's ready to be published as it stands. I outline those concerns below. First though, I would like to strongly encourage the authors to continue developing this manuscript, despite the significant review and publication delays since the first BioRxiv preprint. I have no doubt this will be a valuable piece of work in this important area.

Primary concerns:

1. The authors use three different mechanisms for generating fake data, random number generation, resampling and imputation. These are implemented with the aim of approximating the real data as accurately as possible. It is far from clear to me that any of these strategies reflect strategies scientists would actually use to fake data, however they are as reasonable as any other strategies given the lack of evidence base on this. The way they are used however is where my concern lies.

What motivation would someone have to fake data that reflects the real data and doesn't generate any clear 'result'?!? What the authors have here is a model that detects simulated data with the same characteristics as the current data. I suppose it is possible that scientists might want to fake (simulate) extra samples with similar statistical characteristics as their real data so as to inflate the sample number in their experiments, but it seems far more likely to me that scientists would try to fake data to generate a result.

For this paper, the simplest fake result to add to the data would be a shift in the CNV value to higher or lower values for specific subsets of samples or (more likely) for specific genes within specific subsets of sample. This more realistic test would be simple to implement within the three methods used here. To summarise; in order to be convinced that these methods are useful, I want to see their performance on a real world dataset with a CNV-treatment result in it, with different types of faked data (global up-/down- & specific gene up-down) added to either enhance/deplete the significance of the result, or to add new results to the data. I'd also like to see how models trained on the fake data with these signals in perform; retraining of the models here would probably require a more nuanced investigation for the training in order to avoid training the models just to recognise the up-/down-regulation, rather than the other characteristics of the fake data.

We appreciate the reviewer’s concern on this point and understand the fundamental question to be about whether our methods of making up data are realistic. We offer as a counter point two very high profile papers have been questioned. Both these papers make up data in a manner similar to our method. And the fraud is detected using digit frequencies. Therefore, we feel that our methods are realistic.

The first example, extensively explained here (https://datacolada.org/98), a PNAS paper is discovered to have fabricated half of their dataset. Moreover, and this is very important, the data can be observed to be fabricated based partially on a severe anomaly in the digit frequencies. The paper doubled the dataset by duplicating data points and introducing a strong signature discovered in digit frequencies.

The second and very recent example, extensively explained here (http://steamtraen.blogspot.com/2021/07/Some-problems-with-the-data-from-a-Covid-study.html), a COVID study was shown to be fabricated and contained duplicated patients. Again, one of the ways that fabrication was discovered and confirmed is through digit biases.

2. For the model trained on the two decimal digits, the models are essentially being trained to detect data that doesn't obey Benford's-law. The authors haven’t demonstrated that the machine learning models outperform the far simpler process of making the appropriate histogram and fitting a curve based on Benford's law to this and seeing if you get a decent fit (with a KS test, for example). It's possible that the ML models outperform this simple test, but the authors need to do this comparison to motivate the use of the more complex and opaque ML algorithms.

We do not claim to be identifying data that simply “doesn't obey Benford's-law”. We refer to our method as “Benford-like digit frequency”, it is inspired by Benford’s Law and takes the idea of digit frequencies but the digit distributions of real data in supplemental figure 3, while reminiscent of it, is not Benford’s distribution. Comparing our Benford-like digit-frequencies to the real distribution of Benford’s law is not a useful exercise. Additionally some of our ML models are no more complex than a KS test. We are not convinced that we need to prove one better than the other. We found a variety of methods that appear to work. 

3. Figures 2 & 3 contain boxplots suggesting that some of the models have zero variation in their performance across different data subsets. In some cases this is because the clarifiers are apparently perfectly good/bad accuracy (which I am deeply suspicious of and seems too good to be true) but in some cases it's perfectly consistent accuracy performance (e.g. Fig 2 panel C). These results seem to be in disagreement with Figure 4 which suggests that the average accuracy performance never reaches 100% for any of the models. Something isn't right here. The authors need to carefully inspect their methods reconcile these figures, and either convincingly justify the perfectly good/bad/consistent performance or (more likely) fix the bug that’s causing these.

The largest x value used in Figure 4 was 17,000 (as opposed to the complete full number of features 17156) which can explain this change of accuracy. Because the analysis of Figure 4 depends on repeatedly randomly sampling a subset of features, using the true full set of features had to be excluded as there is only one way to pick 17156 features from 17156 features. Clarifying detail has been added that 17,000 was the max number of features used and an explanation as to why.

Detail comments:

1. Line 69/70. I think the readers would benefit from adding some clarity on the limitations of Benford's law here. In particular, it's only really valid for data that spans several orders of magnitude, and for data where the upper/lower limits are not tightly bounded.

Clarification added about the importance of spanning orders of magnitude and not being bounded

2. Line 86. I disagree with the statement that "making up data is always wrong"; a bit more nuance is needed here. Firstly, simulating data has a long history of being informative in many areas of science. Secondly, there is a grey area here around imputed data, and particularly the imputation of missing data. It is, for example, commonplace to model a covariate in order to impute missing values in this data, and then to use this covariate data - including the imputed data - in a second model which leads to interpretation. From a certain perspective (my perspective, for example!) this could be seen as 'making up data' that has a direct impact on results/conclusions (depending on the scale of the missing data). This is certainly not widely considered wrong or inappropriate.

We are grateful for pointing out this difficult description in the text. We are certainly not suggesting that imputation is wrong, and the reviewer is correct that it is widely used. Here we use algorithms which perform imputation for the purpose of inventing/fabricating datasets. As stated in the introduction, there is a nuance in the way we categorize this which depends on the author’s intent. We have added some new text to hopefully clarify our thinking.

3. Line 111-113: does this mean that some of the samples are represented twice in the 150 sample dataset, albeit with 10% imputed data? How do we know that the ML models are learning to separate the fake samples from the real based on the imputed data signal, rather than needing both a duplicated sample and the imputed data signal. If you added samples that aren't in the original data, with a 10% imputation, would the performance of the models be as good?

We are again grateful to the reviewer for pointing out this section and how it needs to be clarified. In our method, the samples that are fabricated via imputation are composed of 100% fabricated imputation data. The data was just nullified and imputed in chunks of 10%, repeated 10 times so all the data in the final fake sample was fake and unique to it. Doing it in chunks like this was a necessary adjustment to meet the expectations of missForest (plus it sped up an extremely slow process). Additional detail has been added to this section.

4. Line 166: "Machine learning cannot…" I know what you're getting at, but this is not well worded. I think you want to say something like : "Trained ML models are restricted to data that conform to the model input specifications (i.e. the same number of input features, for example).

Thank you. We have adopted your explanation.

5. Line 168: I think it would be worth noting here that the generalizability of this model comes with a cost - it will only work for data where Benfords-law should be valid, which is certainly not all datasets.

I have added a clarifying statement about this “though it’s effectiveness will still be dependent on the existence of digit-frequency patterns”

6. Line 177: "tiddy-verse" should be "tidyverse"

Fixed

7. Line 233: "six" this should be five - this needs checking throughout the paper since the number of models used has changed through the review process.

Fixed, the other review caught this too.

8. Line 247: "The remaining four models…". I think this region of text has been re-ordered quite a bit during the review process and it doesn't make much sense now since we haven't has the results for the fifth model yet at this point. I think the authors need to give this section a careful read and make sure it flows sensibly.

Yes, some mistakes were made and not caught after our first round of revisions (number mismatch and confusing use of “remaining”). Changes have been made to this paragraph and are highlighted.

9. Line 286: "[29149684]" I think this should be a reference??

Yep, thanks

10. Line 289: "While Benford's law…" The shift to use the decimal point digits rather than the leading digit is necessary because of the constraint that Benford law works (best) for numbers spanning several orders of magnitude. This is not the case for the first digit in the CNV data, but this value is usually a non-zero value so the first and second digits necessarily span orders of magnitude. This is a dataset specific approach though. in a dataset comprised mainly of numbers between 0 and 0.09, you would need to use the third and fourth decimal point digits. This would be work illuminating here.

These details have been added.

11. Line 301-307: "Machine learning typically…" repetition of previous text and discussion. I think this can be removed.

Agreed and removed. Brevity is better

12. Line 339. I'm not very surprised at the reasonable performance with as few as 10 genes here. 10 genes x 75 samples = 750 datapoints. This is plenty to build a histogram to compare with the Benfords Law curve (the equivalent of which is what the ML models are learning to do) .

13. Line 353: I think "per data point" should be "per sample" here.

Changed

14. Figure 2: I can't really see the details of this figure well - the resolution is quite low in the PDF embedding. It would be useful to explain the components of the box plot (median, quartiles, indents, etc) for those not familiar with boxplots.

Sorry the resolution was low, our figures were all saved and submitted with fairly high resolution. I assume some resolution was lost in the PDF embedding process but hope that is not the case in the actual publication. If the next figures still look fuzzy you can find all the actually .png and .tiff files properly labeled and in the Figures directory on the Github repo: https://github.com/MSBradshaw/FakeData/tree/master/Figures

15. Supp. Fig. 3. This figure is really useful (I'd put it in the main paper) but it's a nightmare to read because it's very busy. I suggest that the authors split the figure into four facet panels with one dataset per panel.

Supplemental Figure 3 has been split in two a 4 panel plot

---

## [Editor Report · Decision Letter 2]

10 Nov 2021

Detecting fabrication in large-scale molecular omics data

PONE-D-20-32745R2

Dear Dr. Bradshaw,

We’re pleased to inform you that your manuscript has been judged scientifically suitable for publication and will be formally accepted for publication once it meets all outstanding technical requirements.

Kind regards,

Frederique Lisacek

Academic Editor

PLOS ONE
---

## [Editor Report · Acceptance letter]

19 Nov 2021

PONE-D-20-32745R2 

Detecting fabrication in large-scale molecular omics data 

Dear Dr. Bradshaw:

I'm pleased to inform you that your manuscript has been deemed suitable for publication in PLOS ONE. Congratulations! Your manuscript is now with our production department. 

Kind regards, 

on behalf of

Dr. Frederique Lisacek 

Academic Editor

PLOS ONE